# Double Trouble: How Microbiome Dysbiosis and Mitochondrial Dysfunction Drive Non-Alcoholic Fatty Liver Disease and Non-Alcoholic Steatohepatitis

**DOI:** 10.3390/biomedicines12030550

**Published:** 2024-02-29

**Authors:** Wesam Bahitham, Siraj Alghamdi, Ibrahim Omer, Ali Alsudais, Ilana Hakeem, Arwa Alghamdi, Reema Abualnaja, Faisal M. Sanai, Alexandre S. Rosado, Consolato M. Sergi

**Affiliations:** 1King Abdullah International Medical Research Center-WR, King Saud bin Abdulaziz University for Health Sciences, Ministry of National Guard for Health Affairs, Riyadh 11426, Saudi Arabia; wesam.bahitham@kaust.edu.sa (W.B.); alghamdi108@ksau-hs.edu.sa (S.A.); omar325@ksau-hs.edu.sa (I.O.); alsudais207@ksau-hs.edu.sa (A.A.); hakeem023@ksau-hs.edu.sa (I.H.); alghamdia001@ksau-hs.edu.sa (A.A.); abualnaja103@ksau-hs.edu.sa (R.A.); 2Bioscience, Biological and Environmental Sciences and Engineering Division (BESE), King Abdullah University of Science and Technology (KAUST), Thuwal 23955, Saudi Arabia; alexandre.rosado@kaust.edu.sa; 3Gastroenterology Unit, Department of Medicine, King Abdulaziz Medical City, Jeddah 21423, Saudi Arabia; sanaifa@ngha.med.sa; 4Anatomic Pathology, Children’s Hospital of Eastern Ontario (CHEO), University of Ottawa, Ottawa, ON K1N 6N5, Canada; 5Laboratory Medicine and Pathology, University of Alberta, Edmonton, AB T6G 2B7, Canada

**Keywords:** liver, steatosis, steatohepatitis, NAFLD, NASH, fibrosis, cirrhosis, cancer

## Abstract

Non-alcoholic fatty liver disease (NAFLD) and non-alcoholic steatohepatitis (NASH) are closely related liver conditions that have become more prevalent globally. This review examines the intricate interplay between microbiome dysbiosis and mitochondrial dysfunction in the development of NAFLD and NASH. The combination of these two factors creates a synergistic situation referred to as “double trouble”, which promotes the accumulation of lipids in the liver and the subsequent progression from simple steatosis (NAFLD) to inflammation (NASH). Microbiome dysbiosis, characterized by changes in the composition of gut microbes and increased intestinal permeability, contributes to the movement of bacterial products into the liver. It triggers metabolic disturbances and has anti-inflammatory effects. Understanding the complex relationship between microbiome dysbiosis and mitochondrial dysfunction in the development of NAFLD and NASH is crucial for advancing innovative therapeutic approaches that target these underlying mechanisms.

## 1. Introduction

The liver, a vital organ within the human body, plays a pivotal role in various physiological processes and possesses a remarkable capacity for self-regeneration. It performs essential tasks such as glycogen, protein, and enzyme synthesis. Additionally, the liver metabolizes harmful toxins through tightly regulated biochemical processes [1,2]. However, multiple conditions have been shown to impact the physiological functioning of the liver. One prominent example is non-alcoholic fatty liver disease (NAFLD). NAFLD encompasses various disorders characterized by the accumulation of excessive fat in the liver among individuals who consume little or no alcohol and have no other identifiable cause for hepatic steatosis (such as viral hepatitis, lipodystrophy, or certain medications) [3]. This excessive fat deposition in the liver leads to inflammation and progressive liver injury through multiple mechanisms [4]. NAFLD has emerged as a significant public health concern and is currently the leading cause of chronic liver disease worldwide [4] and the second leading cause of liver transplantation in the United States [5]. In the United States alone, it is estimated that 80–100 million adults have NAFLD [3,4]. Approximately 20% of individuals with NAFLD in the United States present with non-alcoholic steatohepatitis (NASH), which is a more severe form of the disease and can progress to cirrhosis, hepatocellular carcinoma, and liver-related mortality [6]. Nowadays, NAFLD is classified as the liver manifestation of metabolic syndrome (MS). This condition is characterized by several metabolic abnormalities, including obesity, elevated blood triglyceride (TG) levels, low high-density lipoprotein (HDL) cholesterol levels, and fasting glucose abnormalities. Collectively, these markers contribute to diagnosing and characterizing NAFLD as part of the metabolic syndrome [5,7,8,9]. NAFLD is also associated with cardiovascular complications and malignancies, with cardiovascular disease being the leading cause of mortality in individuals with NAFLD [8]. NAFLD is considered a multiple-hit disorder, with multiple factors contributing to its pathogenesis. The disease is complex and involves various metabolic, environmental, genetic, and microbiological mechanisms [6,10,11,12,13,14,15,16,17]. Pathogenetic factors associated with NAFLD include established factors such as genetic factors (e.g., PNPLA3 gene polymorphisms) [11,13], dietary factors (e.g., over-nutrition, fructose), insulin resistance (IR) [12,14], and adipokines [13,15]. Additionally, potential factors require further validation, including endocrine disruptors [9,16] and dysbiosis of the gut microbiota [14,17]. These factors collectively contribute to the development and progression of NAFLD. Most individuals with NAFLD do not experience noticeable symptoms or may only report nonspecific symptoms, such as fatigue [15,18,19]. As the disease advances to NASH and eventually cirrhosis, the enlargement of the liver may cause a sensation of weight, discomfort, or even pain in the right upper quadrant of the abdomen. However, these symptoms are not specific to NAFLD, making it difficult to detect the disease without the guidance of a healthcare professional who may recommend imaging studies and/or blood tests to assess liver function [15,19,20]. A liver biopsy is considered the gold standard for diagnosing fatty liver disease and assessing its severity. While liver biopsy or imaging methods provide reliable diagnoses, they are challenging to implement on a large scale for screening and monitoring purposes [16]. Therefore, there is a pressing need to identify individuals at substantial risk of NAFLD or individuals who are in the preliminary stages of the disease. Early identification is crucial because lifestyle interventions can potentially reverse the disease when implemented during the initial stages [17,21]. Moreover, the gut microbiome has become a subject of significant interest in searching for potential new and effective diagnostic and treatment options for NAFLD. This interest is primarily based on human observational studies and animal experiments, which have consistently shown alterations in the gut microbial community composition in individuals with NAFLD [18,19,20]. Since obese individuals frequently exhibit an imbalanced microbiome, known as dysbiosis [22], and given that both the content and quantity of diet greatly impact the composition and function of the human microbiota, it is understandable that the pathogenesis of NAFLD and its progression to more severe conditions is complex. The progression is widely recognized to involve multiple factors, including alteration in the gut microbiome community [18,19,20]. Lastly, the life expectancy of individuals with NAFLD is lower compared to the general population. While some cases of NAFLD remain stable and asymptomatic [21], most patients with NAFLD are at risk of dying from cardiovascular events. Additionally, a smaller proportion of patients may experience mortality due to malignancy and cirrhosis [23].

## 2. Gut–Liver Axis: Communication between Gut Microbiota and Liver

The gut–liver axis (GLA) refers to the interconnected relationship between the liver and the gastrointestinal tract, which has both anatomical and functional aspects. This interaction significantly impacts the gut microbiome and the body’s immune system [24,25]. The GLA involves two crucial components: the intestinal barrier and the gut microbiota. Alterations in either of these components can contribute to the acceleration of liver damage [26]. The observed changes in the gut–liver axis include the presence of small intestinal bacteria overgrowth (SIBO), dysbiosis (imbalanced gut microbial composition), and an increase in intestinal permeability, commonly known as leaky gut [27]. Dysbiosis refers to an imbalance between the normal and pathogenic gut microbiota populations. This imbalance can lead to the secretion of toxins into the liver through the portal vein, facilitated by factors that increase the permeability of the intestinal barrier. In the context of NAFLD, the role of the gut–liver axis (GLA) in this process is significant. The GLA, which encompasses the border of the intestinal lining, is responsible for regulating the translocation of products generated by the gut microbiota [28]. Disruptions in homeostasis can lead to the breakdown of the intestinal barrier, promoting the phenomenon of “bacterial translocation” [29].

Consequently, dysbiosis at an early stage can result in an elevated translocation of toxins and inflammatory substances, thereby affecting the immune response in the body. This can lead to the release of pro-inflammatory cytokines [30]. Another interesting description of the gut–liver axis that recently caught attention due to the addition of another organ to the complex communication process is the gut–liver–brain axis. The gut, brain, and liver have intricate interactions with each other. Intestinal signals can activate lipid-sensitive signals in the hypothalamus through the vagal afferent nerves, regulating food intake [31].

Conversely, the brain inhibits hepatic glucose production to prevent obesity, while the liver inhibits hepatic glucose output through the insulin signaling pathway, reducing brain glucose uptake and neuronal cell activity [32]. On the one hand, gut dysbiosis resulting from a high-fat or high-sugar diet increases intestinal permeability and triggers the production of inflammatory cytokines in colonic epithelial cells [33]. This alteration disrupts gut–brain communication via the vagal afferent nerve. Persistent inflammation activates the gut–vagal afferent nerve, leading to a cascade of sickness-related behaviors in the liver, such as insulin sensitivity and hepatic steatosis [34]. On the other hand, glucagon-like peptide-1 (GLP-1) and its receptor (GLP-1R) play a crucial role in the gut–brain–liver axis. They primarily promote glucose-dependent insulin secretion and reduce body weight through various mechanisms [35]. Notably, the gut microbiota is intricately linked to GLP-1 secretion during the development of NAFLD. Studies have indicated that dysbiosis of the gut microbiota and its metabolites can stimulate GLP-1 secretion via the GPR41/43 pathway, leading to fat accumulation and the development of NAFLD [36]. Short-chain fatty acids (SCFAs), the primary metabolites produced by the gut microbiota, can mimic vagus nerve signaling and regulate neurotransmitters such as serotonin, dopamine, and noradrenaline, influencing brain function [37].

Furthermore, SCFAs play a role in modulating the host’s appetite and food intake, leading to the release of GLP-1 and peptide YY. This occurs through their interaction with G-coupled proteins expressed by enteroendocrine cells, activating the gut–brain–liver axis [38]. These processes contribute to the development of NAFLD. 

## 3. Importance of Microbiome and Mitochondrial Alterations

In normal conditions, a well-balanced gut microbiota is advantageous for human health as it plays a crucial role in maintaining metabolic equilibrium, including the regulation of energy metabolites, lipid metabolism, and glucose metabolism [39]. On the other hand, an imbalance in the gut microbiota disrupts intestinal permeability and disturbs microbiota homeostasis. This disruption facilitates the movement of bacterial endotoxins and other metabolites into the bloodstream, impacting the overall functioning of the human body [40]. Moreover, an analysis of the composition and function of the human gut microbiota revealed that a strong correlation exists with various liver diseases. These diseases include hepatic steatosis, NAFLD, alcohol-associated liver disease (ALD), alcohol-associated hepatitis (AH), chronic cholestatic conditions like primary biliary cholangitis and primary sclerosing cholangitis, autoimmune liver disease, complications associated with cirrhosis and portal hypertension such as hepatic encephalopathy (HE), infections like spontaneous bacterial peritonitis, and hepatocellular carcinoma [41].

Moreover, in a comprehensive study involving a diverse population ranging from healthy individuals to those with severe liver decompensation, we observed significant alterations in the composition of stool microbial communities. These changes were characterized by decreased potentially beneficial autochthonous taxa, such as *Lachnospiraceae*, *Ruminococcaceae*, and *Clostridiales* XIV. Conversely, there was an overgrowth of potentially pathogenic taxa, including *Staphylococcae*, *Enterobacteriaceae*, and *Enterococcaceae*. These shifts in microbial composition were found to be associated with disease progression and the presence of endotoxemia [42]. When it comes to NASH and NAFLD, there has been a growing focus on investigating the impact of the microbiome on metabolic disorders, specifically in understanding the functional importance of the gut microbiome in the advancement of liver disease associated with these conditions [43]. An imbalance in the composition of the microbiome, known as a dysbiotic microbiome, is frequently observed in individuals who are obese [22], and since obesity is a major risk factor for the development of NALFD, both obesity and NAFLD are linked to an increased abundance of Gram-negative bacterial species within the gut microbiome [43]. Furthermore, the microbial populations found in individuals with NASH have been proposed to exhibit an enhanced capacity for alcohol production [44]. Also, NASH has been linked to alterations in bile acid profiles in both serum and feces. This disruption is believed to be a result of reduced bacterial diversity and the loss of specific gut microbiota members that play a crucial role in the synthesis of secondary bile acids [45]. Moreover, certain members of the upregulated gut microbiota can convert choline into trimethylamine, a compound that has been shown to cause liver damage and contribute to the development of steatohepatitis [46]. Therefore, it has been established that alteration in the gut microbiome plays a critical role in the development and progression of both NASH and NAFLD [47,48]. To better conceptualize the association between the gut microbiota and NAFLD, it can be explained by the following progression:

1. The gut microbiota composition is influenced by an individual’s diet and the use of antibiotics, which can contribute to the development of NAFLD.

2. Metabolites produced by the gut microbiota, such as SCFAs and bile acids (BAs), interact with mitochondrial function, genes, and inflammatory factors, thereby promoting the progression of NAFLD.

3. Imbalances in the gut microbiota led to increased intestinal epithelial barrier permeability, allowing harmful substances like metabolites, lipopolysaccharides (LPS), bacteria, and bacterial DNA to enter the liver.

4. The disruption of the gut microbiota also results in elevated levels of LPS in the blood or liver, triggering hepatic inflammation [49]. The gut microbiota and their metabolites play crucial roles in processes such as mitochondrial biogenesis, metabolism, and modulation of oxidative stress. In recent studies, a strong connection between the microbiota and mitochondria has been extensively elucidated in various diseases [50,51,52,53]. The mitochondria serve as the energy source for the continuous production of adenosine triphosphate (ATP), and they are also the primary site of cellular reactive oxygen species (ROS) generation. Consequently, any changes in mitochondrial function can contribute to the accumulation of fat in the liver, lipid peroxidation, increased oxidative stress in the liver, and insulin resistance (IR) [54,55]. Recent research [56] has revealed that changes in the gut microbiota and its metabolites can trigger the buildup of ROS in mitochondria. This, in turn, causes disturbances in oxidative stress and results in mitochondrial damage. These effects have been observed in the progression from hepatic steatosis (HS) to NASH and, eventually, fibrosis. In cases of obesity, both with and without NAFL characterized by steatosis, there is increased availability of free fatty acids (FFAs). This leads to several effects:FFAs increase the intracellular pool of fatty acyl-CoA (FA-CoA).This stimulates mitochondrial fatty acid oxidation (FAO) and may also enhance activity in the tricarboxylic acid (TCA) cycle and electron transport chain (ETC).The upregulated mitochondrial oxidative capacity initially protects against lipotoxicity-induced insulin resistance and the accumulation of triglycerides (TAGs). Additionally, increased catalase and GPX1 activities help scavenge ROS.However, as steatosis progresses to NASH, continuous excess FFA overload impairs the efficiency of mitochondrial oxidative capacity.Lipotoxic metabolites such as ceramides and diacylglycerols (DAGs) accumulate, leading to insulin resistance along with increased gluconeogenesis (GNG) and de novo lipogenesis (DNL).As antioxidant activity decreases, there is an increase in ROS production, resulting in the oxidation of membrane lipids, proteins, and DNA.This impairs mitochondrial biogenesis and quality control.Activation of c-Jun N-terminal kinase (JNK) and nuclear factor kappa B (NF-κB) pathways occurs.Ongoing oxidative stress, hyperglycemia, and dyslipidemia activate Kupffer cells and stellate cells, leading to inflammation, fibrosis, and disease progression through the release of cytokines like tumor necrosis factor-alpha (TNF-α), interleukin-1 beta (IL-1β), and IL-6.

In summary, the interaction between FFAs, mitochondrial function, oxidative stress, and inflammation is critical in the progression of NAFLD/NASH [57]. In NAFLD, the NADPH oxidase system, specifically NOX2, is a significant source of ROS that generates superoxide radicals. Increased NOX2 activity has been observed in the liver cells of NAFLD patients, leading to the release of NOX2 into the bloodstream and elevated levels of serum sp-NOX2, indicating systemic oxidant stress [58,59]. Another marker of oxidative stress in NAFLD is urinary 8-iso-PGF2 alpha, which is associated with increased lipid peroxidation and oxidative damage. Systemic oxidant stress, as indicated by serum sp-NOX2 and urinary 8-iso-PGF2 alpha, contributes to the progression of NAFLD through various mechanisms [60]. It promotes lipid peroxidation, leading to the accumulation of harmful lipid byproducts in the liver that trigger inflammation and contribute to the development of NASH, a more severe form of NAFLD. Oxidative stress also impairs insulin signaling, exacerbates insulin resistance, and further promotes fat accumulation and liver inflammation. Additionally, systemic oxidant stress activates inflammatory pathways and the release of pro-inflammatory cytokines, contributing to chronic inflammation, liver injury, and fibrosis, which are characteristic features of advanced NAFLD. Another way mitochondrial alteration plays a role in the pathogenesis of NASH/NAFLD is by altering species-dependent metabolite production. Imbalances in the gut microbiota, known as gut dysbiosis, are associated with increased production of SCFAs in the intestines. These SCFAs, such as acetic acid, propionic acid, and butyric acid, promote the transport of monosaccharides, gluconeogenesis, and the synthesis and accumulation of harmful lipids in the liver [61,62]. Specifically, butyric acid has been implicated in promoting hepatic lipid synthesis and subsequent lipotoxicity by modulating the activity of Carbohydrate Response Element Binding Protein (ChREBP) and Sterol Response Element Binding Protein-1 (SREBP-1), which are critical regulators of de novo lipogenesis [63]. Propionic acid, another SCFA, is involved in the pathogenesis of metabolic-associated fatty liver disease (MAFLD) by acting as a direct precursor for lipogenesis [64]. Furthermore, SCFAs have the ability to activate G protein-coupled receptors, with GPR43 being particularly relevant. Activation of GPR43 stimulates hepatic lipogenesis, contributing to the development of NAFLD [65,66].

In summary, the dysbiosis-induced production of SCFAs in the gut can lead to hepatic lipotoxicity through various mechanisms, including modulation of key transcription factors, direct precursor effects, and activation of specific receptors involved in lipogenesis. Another factor contributing to gut microbiota-induced lipotoxicity is the presence of excessive secondary bile acids (SBAs). Intestinal microorganisms, through the conversion of primary bile acids, form SBAs. Primary bile acids play a role in maintaining the balance of the gut microbiota by directly inhibiting pathogenic bacteria and activating the farnesoid X receptor (FXR) [67]. The FXR plays a role in preserving the integrity of the intestinal epithelial barrier and regulates the expression of key transcription factors involved in lipid metabolism, such as SREBP-1c and the liver X receptor (LXR). Activation of FXR leads to a decrease in hepatic lipogenesis [68]. In summary, excessive levels of SBAs, resulting from alterations in the gut microbiota, can disrupt the beneficial effects of primary bile acids, leading to dysregulation of FXR activity. This dysregulation impairs the intestinal epithelial barrier and upregulates the expression of SREBP-1c and LXR, promoting hepatic lipogenesis and contributing to lipotoxicity. 

## 4. Gut Microbiome Composition and Diversity

The human gut microbiota (GM) consists of bacteria, viruses (including phages), fungi, and primitive prokaryotic *Archaea* that reside in the digestive tract. They possess many genes, outnumbering those of their human host by one hundred. The GM plays a crucial role in maintaining human health and influencing the development and progression of diseases. It aids in the breakdown and absorption of dietary nutrients and minerals, produces antimicrobial peptides, ferments dietary fibers into SCFAs, detoxifies harmful substances, and regulates local and systemic endocrine and immunological functions [69]. Among the gut microbiota, bacteria are the predominant group, with the Gram-positive *Firmicutes*, known for producing SCFAs, and the Gram-negative *Bacteroidetes*, which produce hydrogen, being the main phyla. Other phyla present include *Proteobacteria*, *actinobacteria*, and *bifidobacteria*, among others [70]. 

Based on the abundance of specific genera, two primary enterotypes have been identified: Enterotype 1, characterized by the dominance of Bacteroides species, and Enterotype 2, characterized by an abundance of *Prevotella* species. There has also been a proposed third enterotype, referred to as *enterotype H*, which exhibits an abundance of both *Bacteroides* and *Prevotella* species [71]. Increasing evidence suggests that the gut microbiome plays a significant role in the development of NAFLD [18]. Studies conducted in humans have demonstrated distinct differences in the gut microbiota composition between individuals with NAFLD and those without the condition. Furthermore, variations in the gut microbiota have also been observed among individuals at various stages of NAFLD [72,73]. NAFLD often arises from nutritional imbalances, which can stem from excessive caloric intake and inadequate nutrient supply. Furthermore, extensive investigations into the causes of NAFLD associated with overfeeding, overnutrition, and obesity have highlighted the paramount role of gut microbiota alterations in promoting the development of the disease [74]. A high-fat diet (HFD) has been found to significantly increase the abundance of *Firmicutes* and decrease the abundance of *Bacteroidetes* [75]. Another study by Li et al. [76] observed that a high-fat, high-cholesterol (HFHC) diet upregulated the abundance of *Firmicutes* and *Verrucomicrobiota* while downregulating the abundance of *Bacteroidetes*, *Actinobacteria*, and *Proteobacteria*. However, during the progression of NAFLD from NASH to NASH with fibrosis, there was a gradual decrease in the abundance of *Firmicutes*, *Verrucomicrobiota*, and *Actinobacteriota* and a gradual increase in the abundance of Bacteroidetes [75]. In HFD-induced NAFLD mice, the levels of triglycerides (TG) and total cholesterol (TC) in the liver were strongly correlated with the abundance of *Firmicutes* and *Bacteroidetes*. Additionally, imbalances in bacterial microbiota, including *Erysipelotrichaceae*, *Coriobacteriaceae*, *Enterorhabdus*, *Lachnoclostridium*, and *Alistipes*, were associated with alterations in serum lipid levels. The composition of the gut microbiome also varied according to the severity of NAFLD. A cross-sectional analysis involving NAFLD-cirrhosis, NAFLD without advanced fibrosis, and non-NAFLD controls revealed that gut microbiota diversity was lower in NAFLD patients without advanced fibrosis compared to healthy individuals. At the same time, it was higher in NAFLD-cirrhosis patients compared to NAFLD patients without advanced fibrosis. Thus, a decrease in gut microbiota diversity was observed concerning the severity of NAFLD [77]. In patients with NAFLD, there were notable changes in the abundance of specific bacterial taxa. *Streptococcus* spp. abundance increased in NAFLD-cirrhosis and NAFLD without advanced fibrosis, while *Megasphaera* abundance increased only in NAFLD-cirrhosis. On the other hand, the abundance of *Bacillus* and *Lactococcus* increased in patients with NAFLD without advanced fibrosis and in healthy participants. Patients with NAFLD-cirrhosis exhibited an increased abundance of *Enterobacteriaceae*, *Streptococcus*, and *Gallibacterium*, while certain beneficial bacteria like *Faecalibacterium prausnitzii*, *Catenibacterium*, *Rikenellaceae*, *Mogibacterium*, and *Peptostreptococcaceae* were found only in healthy individuals. The composition of the gut microbiome also varied significantly with different severities of hepatic steatosis (HS). Mild steatosis was associated with a decrease in the abundance of *Bacteroidetes*, *Proteobacteria*, *Lentisphaerae*, and *Firmicutes*, while moderate steatosis showed a significant increase in the abundance of *Firmicutes* and *Bacteroidetes*. On the other hand, severe steatosis was characterized by a decrease in the abundance of *Actinobacteria*, *Bacteroidetes*, *Lentisphaerae*, *Firmicutes*, and *Proteobacteria*. The *Firmicutes* bacterium CAG 95 was notably decreased in both severe and moderate steatosis. Some species within the *Firmicutes phylum,* including *Ruminococcus bromii*, *Dorea longicatena*, and *Roseburia* sp. *CAG 182* was found to have regulatory effects on AST, ALT, and uric acid levels, consistent with previous studies [78]. The differentiation in the microbiome between males and females is primarily influenced by sex hormones and sex chromosomes [79]. Previous research has indicated that sex-specific microbiomes may play a crucial role in the development of NAFLD and obesity [80]. For example, the genus *Holdemanella* and family *Erysipelotrichaceae* showed a negative correlation with the android fat ratio in females but a positive correlation in males. Similarly, the family *Ruminococcaceae* exhibited a positive correlation with the gynoid fat ratio only in females. The microbiome species associated with fat distribution differ between males and females, and sometimes, even within the same family and genus, the associations with fat distribution can vary between the sexes [81]. Postmenopausal females with estrogen deficiency have an increased risk of NAFLD progression to fibrosis due to alterations in gut microflora. In male patients with NAFLD, there was a decreasing trend in microbial alpha-diversity, an increased abundance of *Dialister*, *Streptococcus*, and *Bifidobacterium* species, and a decreased abundance of *Phascolarctobacterium*, *Mogibacteriaceae*, *Rikenellaceae*, and *Peptococcaceae*. Conversely, female patients with NAFLD showed an increasing trend in microbial alpha-diversity and the abundance of these taxa, exhibiting an opposite trend compared to males [82]. The gut microbiome plays a significant role in developing NAFLD and NAFLD-HCC. NAFLD-HCC is characterized by an increased abundance of *Proteobacteria* compared to healthy individuals. Patients with NAFLD-HCC showed an increase in *Enterobacteriaceae* and a decrease in *Oscillospiraceae* and *Erysipelotrichaceae* abundances. However, the microbiome signature differed between patients with NAFLD-cirrhosis and NAFLD-HCC. The NAFLD-cirrhosis group exhibited an increased abundance of *Eubacteriaceae*, which was not found in either the NAFLD-HCC or non-NAFLD control groups. Additionally, those with NAFLD-cirrhosis had elevated levels of *Coriobacteriaceae* and lower levels of *Muribaculaceae*, *Odoribacteraceae*, and *Prevotellaceae*. This indicates that an increase in *Helicobacter ganmanii* and a decrease in *Bacteroides* play significant roles in developing NAFLD-HCC [83,84]. A comprehensive examination of 10 cohort studies revealed a positive connection between obesity and HCC. However, among these studies, two did not find any association, and one study even indicated an inverse relationship. A limited number of HCC cases and inconsistent control for confounding factors often hindered these investigations. Nonetheless, the meta-analysis, which includes the most extensive dataset to date, demonstrates that bariatric surgery has a risk-reducing effect on HCC. These findings suggest that the role of bariatric surgery extends beyond weight loss and should be considered for a broader range of individuals [76]. Here, it was observed that the gut microbiomes in individuals with NAFLD were primarily composed of *Firmicutes* and *Bacteroidetes*, with *Firmicutes* being the most dominant. It was also observed that patients with NAFLD had a lower abundance of *Proteobacteria* and *Actinobacteria* in their gut microbiome community. As NAFLD advances from mild/moderate stages to advanced fibrosis, there is a notable shift in the gut microbiota composition. Specifically, there is a statistically significant increase in the abundance of the *Proteobacteria* phylum, while the *Firmicutes* phylum shows a decrease. This change in microbial composition is observed during the progression of the disease. In terms of specific species, within the mild/moderate NAFLD group, *Eubacterium rectale* (with a median relative abundance of 2.5%) and *Bacteroides vulgatus* (with a median relative abundance of 1.7%) were the most prevalent organisms. However, in the advanced fibrosis group, *Bacteroides vulgatus* (with a median relative abundance of 2.2%) and *Escherichia coli* (with a median relative abundance of 1%) were the most abundant species identified. This indicates a shift in the dominant species as NAFLD progresses from mild/moderate stages to advanced fibrosis [18]. In patients with advanced NASH fibrosis, Loomba et al. noticed a reduction in the levels of Gram-positive *Firmicutes* bacteria and an elevation in the levels of Gram-negative *Proteobacteria*, which include *E. coli.* This signifies a pronounced alteration in the microbiota composition, characterized by a discernible transition towards an augmented prevalence of Gram-negative microorganisms. As a result, an imbalance in the microbiota with an abundance of Gram-negative bacteria may potentially play a role in the development of liver fibrosis [18].

## 5. Changes in Microbial Diversity and Mitochondrial Dysfunction

Mitochondrial stress caused by various infections can lead to the disruption of the gut microbiota, leading to dysbiosis. An example of bacteria that can alter mitochondrial-microbiome communication is *E. coli*, which contains colonic acid that increases the fragmentation of mitochondria in the intestines [85]. Antibiotic treatment also contributes to the affected mitochondrial homeostasis by transforming an anaerobic to an aerobic environment. The use of antibiotic treatment also leads to a reduction in bacteria that produce butyrate. Consequently, nitric oxide synthase gene expression increases, releasing nitric oxide (NO). The most crucial function conducted by NO is the transformation of glucose into gluconic acid and lactose into mucic acid through the hydrolysis of the glycosidic bond between the glucose and galactose subunits of the carbohydrate. This conversion aids in developing an environment that allows harmful bacteria such as *Salmonella typhi* to reside. The outer membrane of Gram-negative bacteria contains lipopolysaccharides, which could also contribute to the swelling of mitochondria, alter the mitochondrial metabolism, and damage the epithelial cells (Figure 1). 

## 6. Dysbiosis and Its Role in Liver Disease

Dysbiosis refers to a disruption of the symbiotic relationship between the microbiota and the host, and it can contribute to various chronic diseases both within and outside the gut. These diseases include obesity, malnutrition, neurological disorders, inflammatory bowel disease, diabetes mellitus, metabolic syndrome, atherosclerosis, cancer, and liver disease. Several factors can influence the composition of the microbiota and increase the risk of dysbiosis. These factors include diet, environmental factors, stress, aging, broad-spectrum antibiotic use, and genetic factors [86,87,88,89,90,91,92,93,94,95]. Dysbiosis is characterized not only by changes in the qualitative and quantitative aspects of the microbiota but also by shifts in the production of various metabolites by the bacteria. Dysbiosis can lead to increased intestinal permeability, loss of epithelial integrity, and weakened mucus-associated defense. As a result, viable bacteria, microbial products, and host–bacteria interactions can influence normal physiology and disease susceptibility. These influences can occur locally, signaling to different cell populations of the intestinal mucosa and distant organs, including the liver. In the liver, bacterial products can directly affect hepatocytes or cells of the immune system, such as Kupffer cells or stellate cells. Recognition of pathogen-associated molecular patterns (PAMPs) and damage-associated molecular patterns (DAMPs) through TLRs induces pro-inflammatory signals and may also affect apoptosis. Thus, gut-derived factors and alterations in microbial input can influence hepatic inflammation and injury during liver disease. Advancements in metabolomics and metagenomics have helped shed light on the mechanisms through which dysbiosis and altered metabolic output in the gut affect liver diseases. Increased intestinal permeability allows bacterial products and metabolites to cross the epithelial barrier and reach the liver through the portal vein, potentially triggering an inflammatory response. Examples of bacterial metabolites that have been implicated in disease development include ethanol (produced by the intestinal microbiome) in obesity and NAFLD, short-chain fatty acids derived from dietary fibers, secondary bile acids (BA), trimethylamine from dietary choline during NAFLD, and acetaldehyde during alcoholic liver disease [86,87,88,89,90,91,92,93,94,95]. The role of the gut microbiota in influencing health and disease is becoming more recognized. Studies have shown that the human gut microbiota plays a significant role in host metabolism. This understanding stems from initial observations that germ-free (GF) mice have lower levels of body fat, a characteristic that can be reversed when these mice are colonized with a normal gut microbiota [96]. Emerging evidence from both preclinical and clinical research indicates that the gut microbiota plays a significant role in the development of NAFLD. This involvement primarily occurs through its contribution to obesity, metabolic changes such as insulin resistance, and the promotion of liver inflammation. Additionally, certain bacterial byproducts, including ethanol, can exhibit hepatotoxic effects by stimulating Kupffer cells to produce and release nitric acid and cytokines [97]. Multiple pre-clinical and clinical studies have highlighted the key role of gut microbiota in NAFLD pathogenesis through its impact on obesity, metabolic alterations, and liver inflammation. The composition of the gut microbiota, including the abundance of specific microorganisms such as *Akkermansia muciniphila*, may have implications for the development and progression of NAFLD. Several studies conducted over a decade ago demonstrated the significant influence of the gut microbiota on weight gain and obesity. Germ-free mice were found to gain 42% less weight than mice with gut microbiota, even when consuming more calories. Transplanting the cecal microbiota from normal mice into germ-free mice resulted in a 57% increase in weight gain. Furthermore, germ-free mice could not gain weight even on a high-calorie diet. In another study, gut microbiota transplanted from obese mice led to greater fat gain in recipient mice compared to microbiota from lean mice. This suggested that obesity-associated gut microbiota extracted more energy from the diet by digesting indigestible polysaccharides into SCFAs. Similar findings were observed in human studies, with obese individuals having higher concentrations of short-chain fatty acids in their feces compared to lean individuals. Specific gut microbiota composition has been associated with obesity and subsequent NAFLD development. Obese mice had lower levels of *Bacteroidetes* and higher levels of *Firmicutes* and *Archaea* compared to lean controls. Similar alterations in the *Bacteroidetes*/*Firmicutes* ratio were observed in obese humans. *Enterotype 1*, characterized by the consumption of animal proteins and saturated fat, was associated with obesity, while *Enterotype 2* was associated with a diet high in carbohydrates. However, the recently discovered bacterium *A. muciniphila* has been associated with a non-obese phenotype. Low fecal concentrations of *A. muciniphila* were found in pregnant women who gained excess weight during pregnancy and in obese and overweight preschool children. Conversely, *A. muciniphila* counts were increased in obese mice that underwent gastric bypass surgery. Experimental models have shown that *A. muciniphila* modulates weight gain, type 2 diabetes, and NAFLD. Administration of *A. muciniphila* in high-fat diet-fed mice induced Treg cells in adipose tissue, reducing inflammation and improving glucose tolerance. Similar improvements in insulin resistance, adipose tissue inflammation, fat gain, and endotoxemia were observed in mice with type 2 diabetes following *A. muciniphila* administration. Overall, these studies highlight the vital role of the gut microbiota in NAFLD pathogenesis through its impact on obesity, metabolic alterations, and liver inflammation. The composition of the gut microbiota, including the abundance of specific microorganisms such as *A. muciniphila*, may have implications for the development and progression of NAFLD [96,98,99,100]. Endogenous ethanol is a byproduct of certain gut microbiota species and is absorbed into the bloodstream and transported to the liver through the portal vein. In the liver, alcohol dehydrogenase enzymes metabolize ethanol, resulting in the formation of acetate and acetaldehyde. Acetate can contribute to fatty acid synthesis, while acetaldehyde produces reactive oxygen species, leading to oxidative stress. This process contributes to the accumulation of triglycerides in the liver and fulfills both steps of the “two hits” hypothesis for NAFLD development. Elevated levels of ethanol have been found in obese patients and even in non-alcohol-consuming children with NASH, suggesting its role in the development of NAFLD/NASH. Moreover, studies have shown increased expression of alcohol-metabolizing enzymes such as alcohol dehydrogenase, catalase, and aldehyde dehydrogenase in NASH livers. Ethanol may also promote NAFLD by increasing the permeability of the gut mucosa, which can lead to endotoxemia, a condition characterized by the presence of endotoxins in the bloodstream. In summary, endogenous ethanol produced by the gut microbiota can contribute to NAFLD development through its metabolism in the liver, triglyceride accumulation, oxidative stress, and promotion of gut mucosal permeability and endotoxemia [94,101,102]. Endotoxin is a component of the cell membrane of Gram-negative bacteria. The active endotoxin component is called lipopolysaccharide, which binds to the LPS-binding protein and CD14 receptor to form a complex. This complex interacts with Toll-like receptors (TLRs) and triggers an inflammatory cascade. Genetically obese mice develop steatohepatitis (inflammation and fat accumulation in the liver) when low doses of LPS are infused into their bodies. In NAFLD mice, injection of LPS further promotes liver injury by enhancing the production of proinflammatory cytokines. A high-fat diet also leads to increased circulating LPS in rodents with diet-induced NAFLD. Human studies have shown that individuals with NAFLD have significantly higher circulating endotoxin levels than healthy controls. These elevated endotoxin levels are particularly pronounced in the preliminary stages of fibrosis. Activation of Toll-like receptors leads to the translocation of NF-κB (a transcription factor) into the nucleus, resulting in the transcription of proinflammatory genes such as TNF-α, IL-1β, IL-6, and IL-12. IL-1β promotes the accumulation of triglycerides in hepatocytes by enhancing the activity of diacylglycerol transferase, an enzyme that converts diglycerides into triglycerides. On the other hand, TNF-α inhibits insulin receptors and insulin receptor substrate-1, leading to increased levels of circulating insulin and insulin resistance. This insulin resistance facilitates the entry of fatty acids derived from adipose tissue into the liver. In summary, endotoxin (LPS) derived from Gram-negative bacteria activates Toll-like receptors and triggers an inflammatory response in NAFLD. This inflammation is associated with the production of proinflammatory cytokines and the promotion of triglyceride accumulation in hepatocytes. Additionally, the activation of proinflammatory genes and the development of insulin resistance contribute to the progression of NAFLD [103,104,105,106,107,108,109]. Choline, an essential component of cell membranes, plays a vital role in lipid transport from the liver. The gut microbiota regulates choline metabolism by producing enzymes that convert choline into methylamines. When the liver absorbs these methylamines, they have the potential to induce inflammation. Studies using a mouse model of high-fat diet-induced steatosis have shown that there is a decrease in circulating phosphatidylcholine (a form of choline) and an increase in the excretion of choline metabolites in urine. These findings support the presence of a gut microbiota phenotype that leads to choline deficiency and contributes to liver injury. Choline deficiency contributes to the accumulation of triglycerides in the liver and a decrease in the liver’s secretion of very-low-density lipoprotein (VLDL). Animal studies have demonstrated that a choline-deficient diet can result in liver steatosis, which is reversible upon choline supplementation. It is important to note that the data regarding choline deficiency and its association with NAFLD development are derived from animal models and choline-deficient conditions, which may not fully represent the complexity of NAFLD in humans. However, decreased choline levels and increased levels of toxic choline metabolites may represent a potential mechanism through which gut microbiota-mediated choline deficiency contributes to the development of NAFLD [94,110,111,112,113]. 

## 7. Dysbiosis—Driven Inflammation and Mitochondrial Responses

Gut microbiotas play significant roles in vital functions such as maintaining homeostasis, preventing pathogen colonization, producing vitamins, and maintaining a mature immune system. Gut microbiota disruption is highly related to environmental factors such as sex, diet, antibiotic use, and medications. Mitochondria is the main source of energy in the human body and plays a significant role in intestinal homeostasis. Mitochondria is a sensitive organelle that responds to environmental alterations and energy requirements. According to endosymbiosis theory, mitochondria originate from the fusion of archaebacteria and rickettsia alpha-proteobacterium. Mitochondria generate energy via oxidative phosphorylation (OXPHOS). Mitochondria that reside in the gut provide a hypoxic environment that allows obligate anaerobes to remain in the gut to maintain their homeostasis. SCFAs give energy to the epithelial cells of the colon, which affects the metabolism of mitochondria [85]. BA can influence mitochondrial metabolism. *Bifidobacterium* and *Bacteroides* are the main gut microbiota that transform conjugated BA into secondary BA. Secondary BAs control transcription factors that act on carbohydrate and lipid metabolism, which are regulated by the mitochondria [114]. Moreover, peroxisome proliferator-activated receptor-gamma coactivator (PGC)-1alpha upregulation increases oxidative phosphorylation activity. For example, butyrate, a short-chain fatty acid, is metabolized by the colon to produce NADH, which participates in the process of OXPHOS. Furthermore, butyrate could additionally up-regulate uncoupling protein 2 (UCP2) of the mitochondria, leading to decreased production of ROS [114,115]. Moreover, mitochondria and gut microbiome are considered highly dynamic functionally related entities with variations between individuals and within the individual body. It has become increasingly evident that the composition and activity of the gut microbiota profoundly affect human physiology, including immune function and inflammation [116]. Pro-inflammatory cytokines play a pivotal role in initiating and amplifying the inflammatory cascade. They are involved in the recruitment and activation of immune cells, vascular permeability regulation, and tissue damage induction [117]. Emerging evidence suggests that the gut microbiota influences the production and release of proinflammatory cytokines, thereby impacting the immune response and contributing to the development and progression of inflammatory diseases [118]. One of the mechanisms through which dysbiosis influences liver inflammation involves increased intestinal permeability and the subsequent translocation of microbial products, such as lipopolysaccharides, into the liver [30]. Hepatic Kupffer cells interact with lipopolysaccharides via Toll-like receptor 4 (TLR-4) and activate intracellular signaling pathways, leading to the release of pro-inflammatory cytokines such as tumor necrosis factor-alpha (TNF-α), interleukin-1beta (IL-1β), and IL-6 [119]. 

Furthermore, the metabolic activities of the gut microbiota can affect pro-inflammatory cytokine production. The gut microbiota ferments dietary fibers and complex carbohydrates, producing SCFAs as byproducts. SCFAs, such as acetate, propionate, and butyrate, have immunomodulatory effects and can influence the production of pro-inflammatory cytokines [120]. For instance, butyrate has been shown to inhibit the production of pro-inflammatory cytokines while promoting the release of anti-inflammatory cytokines, contributing to maintaining immune homeostasis. The dysbiosis-induced alterations in the gut microbiota composition and function can disrupt these regulatory mechanisms, leading to an imbalanced production of pro-inflammatory cytokines and chronic inflammation [121,122]. 

Understanding the interactions between the gut microbiota and pro-inflammatory cytokines is important for unraveling the mechanisms underlying immune dysregulation and inflammation-associated diseases. Targeting the gut microbiota through interventions such as probiotics, prebiotics, and fecal microbiota transplantation (FMT) holds promise as a therapeutic approach to modulate pro-inflammatory cytokine production and mitigate inflammation-driven pathologies [123].

## 8. Microbial Metabolites and Their Impact on Liver Health

Changes in the gut microbiota can lead to alterations in metabolites within the systemic circulation, shedding light on the underlying mechanisms of NAFLD. Specific metabolite signatures that are associated with distinct stages of NAFLD have been identified [78].

## 9. N, N, N-trimethyl-5-aminovaleric Acid

Researchers have identified a novel metabolite called N, N, N-trimethyl-5-aminovaleric acid (TMAVA) that can be useful in characterizing the different severities of hepatic steatosis (HS) [124]. It has been found that plasma trimethyl lysine (TML) serves as a precursor for TMAVA, and its metabolism into TMAVA is promoted by *Enterococcus faecalis* and *Pseudomonas aeruginosa* [125]. Interestingly, elevated levels of TML have been observed in patients with steatosis. In a clinical trial, the plasma level of TMAVA was found to be positively associated with the abundance of *Bacteroides stercoris*, *B. uniformis*, and *Parabacteroides distasonis*, while negatively associated with the abundance of *Prevotella copri* [126]. However, the combined metabolic activators (CMAs) were found to significantly decrease TMAVA levels. On the other hand, TMAVA can bind to and inhibit the expression of g-butyrobetaine hydroxylase (BBOX), leading to a decrease in carnitine synthesis [127]. This suggests that TMAVA engages in energy production and conversion, as well as the metabolism and transport of carbohydrates and lipids in the liver. Therefore, TMAVA holds potential as a metabolite signature for predicting NAFLD. Microbial metabolites influence mitochondrial metabolism in several ways, which include the production of SCFAs, secondary BAs, amino acid derivatives, and tryptophan metabolites, as well as other mechanisms thought to have an influential impact.

## 10. Short-Chain Fatty Acids (SCFAs)

SCFAs are a type of bacterial metabolite produced through the fermentation of indigestible fibers in the colon [128]. Numerous studies have highlighted the role of abnormal SCFA levels in the progression of NAFLD [20,84,129,130,131,132]. SCFAs can disrupt the integrity of the intestinal barrier, leading to increased translocation of lipopolysaccharide and elevated liver endotoxemia, thereby promoting the development of NAFLD [132]. Among the SCFAs, butyrate and propionate are the primary components. They could reduce gut inflammation and enhance gut barrier integrity, limiting LPS translocation [133]. Liu et al. [132] significantly reduced butyrate levels in female patients with NAFLD and ovariectomized (OVX) mice. Butyrate was also found to be positively correlated with regulatory T cells (Tregs) and effector IL-10 while negatively correlated with cytotoxic CD8 T-cells in individuals with NAFLD-HCC [84]. Previous studies have indicated that SCFAs can directly influence T-cell immunity through their effects on the gut microbiota [134,135]. Furthermore, supplementation with a high-fiber diet can increase SCFA levels, particularly butyrate, which promotes hepatocyte proliferation [131,136]. Butyrate, nicotinate, and 2-oxoglutarate positively regulate hepatic oxidative phosphorylation and negatively regulate triglyceride content through oxidative metabolism. The intermediates of SCFAs, such as oxaloacetate and acetyl phosphate, are also increased in patients with NAFLD-HCC [84]. Specific bacterial species are strongly associated with the production of SCFAs. For instance, *F. prausnitzii* has been shown to induce apoptosis by regulating mitochondrial death, ROS, and the caspase pathway during the progression from NAFLD to NASH through SCFA production [137]. The levels of SCFAs were also found to be positively dependent on *Peptococcus* and *Romboutsia* while negatively dependent on *Ruminiclostridiun-6* and *Muribaculum* [129]. SCFAs and these bacterial species positively regulated total cholesterol (TC) levels, leptin, and body weight in female participants. Other SCFAs, such as acetate and propionate, may also be associated with bacteria such as *Olivibacter*, *Clostridium*, and *Dysgonomonas* [138]. Acetate and propionate are the main products of the *Bacteroidetes* phylum, and butyrate is mainly produced by the *Firmicutes* phylum. As an energy precursor, SCFAs are implicated in the pathogenesis of NAFLD because of their possible contribution to obesity. The first evidence regarding SCFAs was from a Turnbaugh et al. study showing that the cecum of obese mice has an increased concentration of SCFAs and that transplantation of germ-free mice with the gut microbiome of obese mice caused greater fat gain than transplants from lean animals [139]. In humans, increased production of SCFAs by the gut microbiota was also observed in overweight and obese people, compared to lean subjects [100].

## 11. Bile Acids

Apart from SCFAs, BAs also regulate inflammation associated with hepatic steatosis (HS) by interacting with their respective receptors as agonists or antagonists [133,140,141]. Primary BAs, such as taurocholic acid (TCA), tauroursodeoxycholic acid (TUDCA), glycocholic acid (GCA), and taurochenodeoxycholic acid (TCDCA), were found to be elevated in patients with NASH and mice fed a high-fat, high-cholesterol (HFHC) diet. These primary BAs have been identified as critical metabolites that impact the accumulation of hepatic lipids and inflammation. Increased abundance of *M. schaedleri*, *Roseburia*, and *H. ganmanii* was associated with elevated levels of TUDCA, TCDCA, TCA, and GCA, while decreased abundance of *A. muciniphila* due to the HFHC diet led to increased TCDCA and TUDCA levels [140]. Additionally, an increased abundance of *Anaerotruncus* due to the HFHC diet resulted in a depletion of indolepropionic acid (IPA) [141]. Other bacteria, such as *Roseburia intestinalis*, *P. distasonis*, *Bacteroides vulgatus,* and *B. uniformis,* are also involved in the secondary BA metabolism pathway [78]. In participants with NAFLD, primary BA levels were negatively associated with the abundance of *R. bromii*, a species known to benefit human health. Furthermore, the enrichment of *Bilophila wadsworthia* led to BA dysmetabolism, inflammation, and intestinal barrier dysfunction, contributing to higher glucose dysmetabolism and hepatic steatosis [43]. Therefore, *Bifidobacterium* and *Bacteroides*, which are predominant gut microbiota species, are also involved in BA metabolism in HFHC-induced NAFLD [140]. These species can prevent the conversion of taurine- and glycine-conjugated BAs into their unconjugated free forms [142]. Additionally, BAs play an important role in shaping gut microbiome membership, which is a result of these BAs contributing to the prevention of intestinal bacterial growth, both directly through the membrane-damaging effects and indirectly through the induction of antimicrobial protein expression [143]. Moreover, human studies have noted elevated total serum BAs in adults with NAFLD [144,145,146]. Overall, previous research suggests that exploring treatment strategies for NAFLD may involve reversing impaired BA metabolism to prevent the development of NAFLD-HCC.

## 12. Other Microbiome-Specific Metabolites

The development of NAFLD is also influenced by the production of FAs by specific bacterial species, such as *Firmicutes bacterium CAG 95* and *Firmicutes bacterium CAG 110* [78]. The expression of key hepatic genes involved in FA synthesis, including SREBP1, PPAR-g, FAS, and CHREB, is altered in high-fat diet (HFD)-fed mice and individuals with reduced estrogen levels, contributing to NAFLD [147]. Hepatic lipid accumulation contributes to the uptake of circulating FAs and a decrease in the rate of FA oxidation and secretion [138]. Butyrate, for example, can inhibit lipid synthesis, enhance FA β-oxidation by reversing PPAR-α activation, and reduce the levels of nuclear factor-kappa B, which has also been observed in NAFLD-OVX mice [62,132,148,149]. A newly identified amino acid called 3-(4-hydroxyphenyl) lactate also engages in tyrosine metabolism in NAFLD [150]. Interestingly, circulating 3-(4-hydroxyphenyl) lactate can be produced by *E. coli,* which also produces hydroxyphenyl lactate in vitro [151]. Moreover, members of the *Firmicutes*, *Bacteroidetes*, and *Proteobacteria* phyla can produce 3-(4-hydroxyphenyllactate and phenyl lactate) in NAFLD. Other dysfunctional metabolites are also associated with specific bacterial abundances. For example, carnosine, nicotinate, methylamine, trimethylamine, and arabinose have been linked to the abundance of *Bacteroides* in HFD-induced NAFLD, while *Olivibacter*, *Clostridium*, and *Dysgonomonas* have been correlated with acetate and propionate levels [138]. Although bacterial products are often linked to negative effects, specific bacterial metabolites, for instance, indole, have been discovered to have beneficial effects on the host. These metabolites can modulate intestinal immune responses and influence epithelial integrity, thereby promoting positive outcomes for the host [152,153,154]. Multiple studies have demonstrated that indole and its derivatives have an impact on host physiology through various molecular mechanisms. These compounds play a role in maintaining the balance and stability of both the intestinal and systemic environments by regulating the communication between the microbiota and the host’s innate immune system. Various indole derivatives have been identified as ligands for the aryl hydrocarbon receptor (AhR), a cytosolic transcription factor expressed by immune cells regulating intestinal immune homeostasis [155]. AhR plays a role in antimicrobial defense by inducing the expression of interleukin-22 (IL-22) [156]. IL-22, in turn, regulates the microbial composition and enhances antimicrobial defense by promoting the production of antimicrobial proteins like regenerating islet-derived protein three gamma (REG3G) [157,158]. Additionally, AhR has anti-inflammatory effects and influences the development of intraepithelial lymphocytes and innate lymphoid cells, which play crucial roles in defending against invading pathogens and maintaining gut homeostasis [159,160,161]. The activities of AhR help promote the dominance of commensal bacteria over pathogenic bacteria in the gut microbiota, preventing dysbiosis [162]. Indole has been shown in both in vitro and in vivo studies to benefit the gut by enhancing the integrity of epithelial tight junctions [163,164]. Studies have also demonstrated that a high-fat diet can lead to a depletion of tryptamine and indole-3-acetic acid, which are microbiota-dependent metabolites, in the liver and cecum of mice [165]. Both metabolites have been found to ameliorate lipopolysaccharide-induced cytokine production by macrophages and the infiltration of immune cells via chemokine signaling. Indole-3-acetic acid has also been shown to reduce inflammatory gene expression in lipid-loaded hepatocytes in an AhR-dependent manner [166]. These findings suggest that, in addition to their immune-regulatory effects in the gut, indole derivatives may directly exert protective functions on the liver during inflammatory conditions such as those observed in NASH. Amino acid homeostasis is influenced by the gut microbiome, in part due to the biosynthesis and metabolism of aromatic amino acids (AAAs) and branched-chain amino acids (BCAAs) [167]. Several cohort studies identified elevated serum BCAA as a potential biomarker for insulin resistance [167]. In a cohort study conducted by Hoyles et al., women with NAFLD had significant alterations in the gut metagenome, including differences in BCAA and AAA pathways, as well as in the serum metabolome [19]. This study provides proof of how integrated analyses in human subjects can facilitate the identification of microbial-driven mechanistic pathways in NAFLD.

## 13. Role of Gut-Derived Signaling Molecules in Shaping Mitochondrial Health

Furthermore, several pathogenic bacteria have been identified to cause damage to the mitochondria of the intestinal epithelial cells by increasing oxygen content and disrupting the microbiome environment. These pathogenic bacteria include *Salmonella typhi*, which uses its virulence factors to induce an inflammatory response that increases the oxygenation of epithelial cells. Oxygen is then distributed through the intestinal cavity to inhibit *Clostridium* spp. growth and reduce butyrate concentration. As a result, the mitochondrial OXPHOS pathway is disrupted. Another example of a pathogenic bacteria that utilizes its virulence factors to damage the mitochondria is *Citrobacter* spp. It starts a cascade of epithelial tissue repair followed by stem cell differentiation and cell proliferation, which increase oxygenation and destroy the intestinal mucosal barrier. On the other hand, some pathogenic bacteria contribute to mitochondrial destruction by releasing pore-forming toxins. Among these pathogenic bacteria is *Helicobacter pylori*, which releases VacA toxins that affect the inner membrane of the mitochondria. The disruption of the inner mitochondrial membrane leads to hyperpermeability of the outer mitochondrial membrane. As a result, the electron transport chain (ETC) is affected by the changes in the proton gradient, which further affects the OXPHOS of the mitochondria. Another example of bacteria that secrete pore-forming toxins is *Listeria* spp. In fact, *Listeria* secretes *Listeria* hemolysin O (LLO), which increases cellular calcium influx, leading to disruption of mitochondrial membrane potential and OXPHOS disruption. Moreover, some bacteria, such as *S. flexneri* surface protein (IcsA), can also influence the mitochondrial dynamic negatively, which can lead to the Drp1-dependent fission of epithelial cell mitochondria [85]. 

Mitochondria play a critical role in intestinal epithelial cell metabolism, immunity, and cell apoptosis. The mitochondrial OXPHOS process synthesizes ATP. Damage to cell oxidation leads to a reduced OXPHOS process. Hence, reduced energy production is vital for cells to function normally. Reactive oxygen species (ROS) accompany ATP production and play an essential role in the oxidative defense system of cells.

Nevertheless, high levels of ROS can damage mitochondrial DNA and proteins. Moreover, ROS stimulates inflammasome-like receptors (NLRP3), which promote the formation of inflammasomes or increase mitochondrial permeability to activate oxidized mitochondrial DNA (ox-mtDNA), which is leaked to the cytoplasm and activates NLRP3 due to the overload of ROS. PARKIN-PINK1 pathway activation, mitochondrial autophagy, and oxidative stress damage are also consequences of excessive levels of ROS [85]. 

Normally, aerobic glycolysis is converted to OXPHOS after the stem cell differentiation of intestinal epithelial cells. However, any alteration in mitochondrial metabolism due to external or internal factors will affect gut microbiota structure [168]. For example, in inflammatory processes of the mitochondria, epithelial cells will enter apoptotic pathways due to the over-activation of mtUPR. Additionally, the aging of epithelial cells also contributes to the damage to the mitochondrial metabolic pathways [85].

## 14. Interplay between Gut Microbiota and Mitochondrial Function

Alteration of mitochondrial function can induce liver fat deposition, lipid peroxidation, hepatic oxidative stress, and liver insulin resistance (IR) [115]. Recently, alteration of the gut microbiota and its metabolites has been shown to induce the accumulation of ROS in mitochondria and lead to alterations in oxidative stress and mitochondrial damage, which have been described in hepatic steatosis or NASH progression to fibrosis [55,56,169]. In addition to metabolites such as SCFAs and BA, bacteria can also promote crosstalk between the microbiota and mitochondria by directly regulating the expression of cellular genes. Previous investigations have illustrated members of *Bacteroides*, *Firmicutes*, and other bacteria alternating the level of SCFAs [170], which the mitochondria utilize to synthesize energy [159]. For example, applying sodium butyrate can significantly enhance mitochondrial DNA content in HepG2 cells, increase membrane potential function, and ameliorate mitochondrial dysfunction. Parallelly, NaB can elevate the activity of superoxide dismutase (SOD) and glutathione peroxidase (GPX) and deplete the levels of prooxidative NADPH oxidase 2 (NOX2), ROS, and malondialdehyde (MDA). Furthermore, deacetylation of histones may also be regulated by NaB to improve energy metabolism in NAFLD [126]. In addition to SCFAs, BAs can influence mitochondrial energy metabolism and biogenesis. *Bifidobacterium* and *Bacteroides* are the main gut microbiota that transform conjugated bile acid into secondary bile acid during the progression of NAFLD in HFHC-fed or HFD-fed rats [140,142]. Secondary BAs regulate mitochondrial function by controlling transcription factors, including those involved in carbohydrate and lipid metabolism (Figure 2). 

## 15. Combined Impact on NAFLD/NASH Development and Progression

The gut microbiota plays an important role in the progression of NAFLD as well as its more severe form, NASH. The progression to NAFLD and NASH can be summarized in a few proposed theories. First, dysbiosis and gut permeability. Dysbiosis is associated with the progression of NAFLD/NASH, which results in increased gut permeability, allowing harmful substances to enter the bloodstream, which triggers inflammation and liver damage. SCFAs, the product of dietary fibers fermented by gut microbes, have anti-inflammatory properties and can influence lipid metabolism, leading to the progression of NAFLD. In addition, gut microbiota plays a role in BA metabolism, and any alteration in BA composition can affect lipid absorption and metabolism, influencing hepatic steatosis development. Moreover, some gut microbes can produce ethanol as a byproduct of carbohydrate fermentation. This contributes to liver damage similar to the one produced by excessive alcohol intake. Additionally, dysbiosis can lead to an imbalance in pro-inflammatory and anti-inflammatory signals from gut microbes, which could exacerbate hepatic inflammation and contribute to the development of NASH [167]. Overall, any disruption in the host–microbe interaction can lead to several chronic diseases, including alcoholic liver disease and NAFLD. Several animal studies have provided evidence for the role of the gut microbiome in the pathogenesis of NAFLD. Several potential links between the gut microbiome and NAFLD have emerged based on these animal studies. These mechanisms include dysregulation of methylamine metabolism, carbohydrate fermentation and generation of SCFAs, endogenous ethanol production, bile acid metabolism, and amino acid metabolism [167].

## 16. Clinical Evidence of Microbiome Alterations in NAFLD/NASH

The intricate relationship between gut microbiota and lipid absorption is intricate. The causal relationship between dysbiosis and lipid levels has been demonstrated by research suggesting correlating lipid levels when gut microbiota is transplanted from donors with obesity or non-alcoholic fatty liver disease [171]. One study that has relied on mouse models has presented evidence of increased adiposity and body mass upon transplantation of fecal microbiota from obese donors [171]. Another study, also in mouse models, has demonstrated alleviation of high-fat-induced steatohepatitis in mice with transplanted fecal microbiota, further elucidating dysbiosis’s role in the prognostication of disease rather than its occurrence alone [172]. It is also important to consider the important role the gut microbiota plays in the metabolism of fatty acids. One study that has evaluated fatty acid synthesis has demonstrated the role gut flora plays in the desaturation of hepatic acid, mainly via increased acetate production in the gut, which serves as a precursor for C16 and C18 fatty acids in the liver [173].

## 17. Human Studies of Microbiome Profiles in NAFLD/NASH

Multiple studies evaluating the gut microbiome in NAFLD and its subtypes have been conducted in animals and humans. However, it is significant that findings in animals versus humans vary. Multiple studies have also evaluated fecal and serum metabolites that may also be associated with NAFLD (and its subtypes). However, herein, we only report the gut microbiota as reported in human studies conducted (Table 1). 

## 18. Biomarkers and Indicators of Crosstalk for Disease Prognosis

While recent research seems to suggest the collective role of gut microbiota in the progression of NAFLD, it is of particular importance to study species of predominance in patients with NAFLD. *Prevotella copri* is a Gram-negative commensal gut microbe that has been of particular focus in recent research due to its inflammatory signature. It is thought to induce a resistant state via its superoxide reductase. This microbe has also been shown to induce elevations in pro-inflammatory interleukins IL-6, IL-23, and IL-1 [185]. Certain cytokines have been identified as playing a role in the progression of viral hepatitis, ultimately leading to the development of chronic liver disease [185]. Specifically, IL-6, IL-8, IL-10, and IL-23 have been implicated in HBV infection (9, 10), whereas the predominant immune responses associated with the advancement of HCV infection involve IL-10 and IL-12 [185]. This inflammatory status has been shown to be characteristic of advanced fibrosis in NALFD and other inflammatory conditions, specifically rheumatoid arthritis [186]. With *P. copri* in mind, it is also vital to consider the role of carbohydrate metabolism and fat metabolism, namely linoleic acid, which appears to coincide with advanced fibrosis in patients with NAFLD [187]. 

Another prognostic factor that has recently come to light is non-coding RNAs. These makeup most of the human genome and mainly act to stabilize the messenger RNAs post-transcription. They are found in multiple lengths, including microRNAs and long noncoding RNAs. Recent research has shown those particular micro-RNAs, namely miR34, miR-192, miR-375, and miR-122 [188], play the largest role in the prognostication of NAFLD progression as they are often upregulated in the serum and downregulated in the liver in patients with NAFLD. As such, mirR-122 has been shown to show a 7-fold change in serum levels in NASH compared to controls in one study.

Nevertheless, miR-122 appears to undergo deregulation by up to 10 folds in the liver in patients with NASH vs. controls, presenting evidence for its central regulation associated with the progression of NASH [188]. In one study that examined the histopathological features associated with mirR-122, the progression of NAFLD was related to the expression of miR-122 in the serum on serial liver biopsies [188]. Long non-coding RNAs (lncRNAs) have also been shown to play a significant role in the development and progression of fibrosis in NAFLD in animal and human trials, setting up the stage for future use of this data point as a prognostic factor [189]. Nonetheless, it is important to recognize that miRNAs and lncRNAs have a coregulatory role [190].

## 19. Therapeutic Intervention Targeting the Microbiome

Therapeutic interventions targeting the microbiome have gained considerable attention as a promising intervention for improving health outcomes [191]. By modulating the composition and activity of microbial communities, these interventions aim to restore microbial homeostasis and potentially alleviate or prevent a wide range of diseases [192]. Different therapeutic interventions targeting microbiomes include probiotics, prebiotics [118], fecal microbiota transplantation (FMT), and dietary interventions interfering with the action of gut microbiota. 

## 20. Probiotic and Prebiotic for Microbiome Modulation

Despite the alarmingly high prevalence of NAFLD, no pharmacological agent has yet been approved by the Food and Drug Administration (FDA) [147,193,194]. However, in recent years, a growing body of literature has explored the use of pharmacological agents to manipulate the gut microbiota as a potential alternative treatment for NAFLD, specifically targeting the microbiome [195]. The notion of utilizing pharmacological agents to modulate the gut microbiota was derived from introducing probiotics, a combination of live microorganisms intended to restore microbial homeostasis by modulating dysbiosis and promoting immune system regulation [196]. Recent studies have yielded mounting evidence supporting the potential contributions of gut microbiota, particularly members of the *Bifidobacterium* and *Lactobacillus* genera, in various aspects of human health. These probiotic bacteria have shown promise in improving gastrointestinal function, enhancing diabetes treatment outcomes [197,198,199], strengthening the immune system [200,201], and even potentially reducing hospitalization durations [202]. In animal models, the administration of probiotics has demonstrated considerable efficacy in reducing the occurrence of fatty liver disease and ameliorating oxidative stress in mice with NAFLD [203,204]. These findings suggest a potential therapeutic role for probiotics in managing NAFLD.

In addition, prebiotics, which refer to dietary substrates that selectively nourish the growth and activity of beneficial microorganisms within the gut, have gained considerable attention due to their significant health benefits when administered in conjunction with probiotics [205]. Prebiotics have emerged as a critical player in the modulation of dysbiosis, contributing to restoring a healthy microbial balance [206]. In studies conducted on animal models, the administration of prebiotics has exhibited notable effectiveness in mitigating liver lipogenesis in mice afflicted with NAFLD, implying a potential therapeutic role for prebiotics in managing NAFLD [207,208,209].

Nevertheless, empirical evidence has shown that the synergistic combination of prebiotics and probiotics did not yield statistically significant changes in the levels of liver enzymes or liver steatosis in patients diagnosed with NAFLD [118,210]. Despite the theoretical potential of this combined approach to modulate the gut microbiota [206], clinical studies have yet to demonstrate consistent and significant therapeutic effects in terms of liver function and steatosis reduction. These findings highlight the need for further research to elucidate the multifaceted factors that may influence the efficacy of therapy in the context of gut microbiota modulation, including prebiotics and probiotics, for NAFLD. It is crucial to continue investigating the intricate relationship between gut microbiota modulation, prebiotics, probiotics, and NAFLD to develop more targeted and effective therapeutic strategies for this prevalent liver condition.

## 21. Fecal Microbiota Transplantation (FMT)

FMT, also known as fecal microbiome transplantation, is a therapeutic intervention designed to address disruptions in the microbial equilibrium of the gastrointestinal tract by transferring the gut microbiota from a healthy donor to an affected patient [211]. FMT has gained considerable traction recently as a therapeutic approach for a broad range of gastrointestinal and extra-gastrointestinal disorders [212]. The observed efficacy of FMT in these conditions is believed to stem from the intricate interplay between the gut microbiota and various physiological responses [211,212]. Current clinical guidelines strongly advocate the utilization of FMT as a treatment option for patients suffering from recurrent *Clostridium difficile* infection (CDI), a severe, debilitating infection. FMT has demonstrated an impressive success rate exceeding 90% in resolving this challenging condition [213,214,215]. 

Emerging studies have provided insights into the effects of allogeneic (donor) FMT on various aspects of health. Recent research has suggested that allogeneic FMT is linked to a notable increase in intestinal permeability and a substantial reduction in hepatocyte inflammation. In contrast, neither allogeneic nor autologous (own) FMT have demonstrated beneficial changes in terms of insulin resistance or hepatic proton density fat fraction as assessed by MRI [123,216]. Conversely, recent studies have shown promising results regarding the potential benefits of autologous FMT in certain conditions, such as type 1 diabetes and inflammatory bowel disease (IBD) [217,218,219]. Emerging research suggests that allogeneic FMT shows promise for individuals with NAFLD and/or NASH. Allogeneic FMT has been found to induce favorable changes in the intestinal microbiota composition, leading to beneficial alterations in plasma metabolites and markers associated with steatohepatitis [123]. However, further well-designed studies, including randomized controlled trials, are needed to establish the efficacy, safety, and long-term effects of FMT in this patient population. Understanding the underlying mechanisms and conducting comprehensive investigations will be crucial in determining the potential therapeutic implications of FMT for NAFLD and NASH.

## 22. Dietary Interventions and Their Effects on Gut Microbiota

Dietary interventions have emerged as a fundamental modulator of intestinal health, influencing various physiological and pathological processes [220]. Notably, empirical evidence has provided insight into the direct influence of diet on the diversity and composition of the gut microbiota, which, in turn, has been implicated in the pathogenesis of chronic diseases characterized by persistent systemic inflammatory responses, such as type 2 diabetes. Furthermore, a growing body of evidence has provided substantial evidence suggesting that a diet rich in animal-derived and saturated fats possesses the potential to disrupt the delicate equilibrium of the gut microbiota. This perturbation is characterized by elevated levels of lipopolysaccharides and alterations in key metabolites such as trimethylamine-N-oxide (TMAO) and SCFAs [221,222]. Disrupted gut microbiota can compromise the integrity of the intestinal barrier, leading to heightened permeability and increased systemic absorption of LPS. Once LPS enters the systemic circulation, it triggers a cascade of systemic inflammatory reactions and contributes to the development of insulin resistance, a hallmark of conditions such as type 2 diabetes [223].

Conversely, TMAO, an essential amino acid derivative, has emerged as a notable factor in the modulation of the gut microbiota when consumed as part of a diet abundant in TMAO-rich sources. Studies have indicated that a deficiency of TMAO can contribute to compromised intestinal immunity and foster dysbiosis, an imbalance in the composition and function of the gut microbial community. The intricate relationship between TMAO and the gut microbiota highlights the potential significance of TMAO in maintaining intestinal homeostasis and promoting immunity regulation [224]. 

SCFAs are products of the bacterial fermentation of dietary fibers by the gut microbiota [225]. A substantial body of research has consistently demonstrated the association between SCFAs and various health benefits. Specifically, studies have shown that SCFAs are implicated in the enhancement of insulin action [225,226,227], attenuation of inflammatory responses [225,228,229,230,231], promotion of satiety, and even long-term weight loss [225]. Moreover, elevated levels of SCFAs have been found to contribute to reducing lipopolysaccharide translocation, thereby mitigating inflammatory reactions [232]. These compelling findings underscore the importance of dietary factors in modulating gut microbiota composition and their subsequent impact on systemic metabolic health. Further investigations are warranted to elucidate the precise mechanisms underlying these associations and to develop targeted dietary interventions for mitigating chronic low-grade inflammation and metabolic dysfunction.

## 23. Precision Interventions: Integrating Omics Data for Patient-Specific Treatment Strategies

Precision medicine represents a revolution in healthcare, aiming to tailor interventions based on individual biological information [233]. By integrating healthcare data with targeted assays and tests, precision medicine enables the identification and assessment of diseases [234]. While it has revolutionized cancer treatment by matching therapies to specific molecular drivers, its application to complex, multifactorial diseases has been limited due to the scarcity of definitive genetic or protein markers [235]. To address this challenge, precision medicine requires integrating many types of omics data, from genomics and proteomics to metabolomics and phenomics. Effectively analyzing these databases is crucial [235]. Multi-omics strategies, deep phenotyping, and predictive analysis are employed to integrate collective and individualized clinical data with patient-specific multi-omics information, facilitating the development of tailored therapeutic approaches. The ultimate objective of precision medicine is to identify patient subgroups with unique treatment responses or distinct healthcare requirements [236]. The inclusion of food additives and environmental factors in NAFLD studies will also play a major role in stemming out co-etiologic factors and selective therapeutic protocols [237]. Integrating multiple data sources and studying patients longitudinally across different disease stages enables the identification of disease drivers within specific patient clusters, paving the way for precision medicine strategies [236]. Healthcare optimization relies on targeting pre-analytical, analytical, and post-analytical phases of tissue handling properly [238], and the integration of lncRNAs in OMIC studies will be paramount in the near future [239].

## 24. Conclusions

In conclusion, this review highlights the intricate interplay between microbiome dysbiosis, mitochondrial dysfunction, and the emerging field of precision medicine in developing NAFLD and NASH. The combination of microbiome dysbiosis and mitochondrial dysfunction, known as “double trouble”, acts synergistically to promote lipid accumulation in the liver and the progression from simple steatosis to inflammation. Moreover, recent advancements in precision medicine have provided valuable insights into individualized patient care and treatment strategies. Precision medicine recognizes the unique genetic, environmental, and microbial factors contributing to NAFLD and NASH’s development and progression. By integrating genomic and microbial profiling, along with clinical and lifestyle data, precision medicine aims to tailor interventions and therapeutic approaches that specifically target the underlying mechanisms of each patient. Understanding the complex relationship between microbiome dysbiosis, mitochondrial dysfunction, and NAFLD/NASH is crucial for advancing innovative therapeutic strategies based on precision medicine principles. By leveraging this knowledge, researchers and clinicians can develop personalized interventions that address the specific dysfunctions and imbalances in each patient’s microbiome and mitochondrial function. This approach holds promise for improving treatment outcomes, optimizing patient care, and reducing the burden of NAFLD and NASH globally. However, further research is needed to validate and refine the application of precision medicine in NAFLD and NASH. Longitudinal studies and clinical trials are necessary to assess the efficacy and long-term benefits of precision medicine-based interventions in improving patient outcomes and preventing disease progression. By embracing precision medicine, we have the potential to revolutionize the management of NAFLD and NASH and pave the way for personalized therapeutic strategies that target the underlying mechanisms of these complex liver conditions.

## Figures and Tables

**Figure 1 biomedicines-12-00550-f001:**
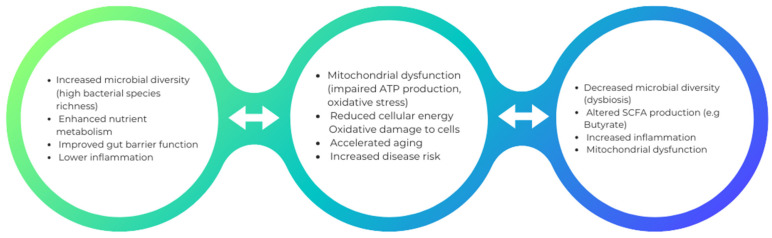
This is an illustrative depiction of how augmented microbial diversity can potentially engender beneficial effects on health, whereas diminished microbial diversity may intricately contribute to mitochondrial dysfunction and its correlated health outcomes.

**Figure 2 biomedicines-12-00550-f002:**

This figure illustrates the dynamic interrelationship between microbial metabolites and mitochondrial metabolism. Microbial metabolites, such as SCFAs and various other bioactive compounds, exert a significant influence on cellular energy production and intercellular signaling processes.

**Table 1 biomedicines-12-00550-t001:** Microbiota microorganisms (by genus) in NAFLD.

No.	Microorganism (Genus)	Status in NAFLD *
1.	*Acidaminococcus*	Increased [19]
2.	*Akkermansia*	Decreased [83]/Increased [19]
3.	*Alistipes*	Decreased [174]
4.	*Allisonella*	Increased [175]
5.	*Anaerococcus*	Increased [176]
6.	*Anaerosporobacter*	Decreased [177]
7.	*Atopobium*	Increased [177]
8.	*Bacillus*	Increased [42]
9.	*Bacteroides*	Increased [178]
10.	*Bifidobacterium*	Increased [179]/Decreased [179,180]
11.	*Blautia*	Increased [176]
12.	*Bradyrhizobium*	Increased [44]
13.	*Clostridium*	Increased [174]
14.	*Coprobacter*	Decreased [19]
15.	*Coprococcus*	Decreased [181]
16.	*Dialister*	Increased [182]
17.	*Dorea*	Increased [180]
18.	*Eggerthella*	Increased [19]
19.	*Enterococcus*	Increased [83]
20.	*Escherichia*	Increased [183]
21.	*Eubacterium*	Decreased [19]
22.	*Faecalibacterium*	Decreased [181]
23.	*Flavonifractor*	Increased [19]
24.	*Haemophilus*	Decreased [19]
25.	*Holdemania*	Decreased [19]
26.	*Klebsiella*	Increased [178]
27.	*Lachnospiraceae incertae sedis*	Increased [183]
28.	*Lactobacillus*	Increased [83]/Decreased [174,181]
29.	*Megasphera*	Increased [42]
30.	*Moryella*	Decreased [177]
31.	*Neisseria*	Decreased [177]
32.	*Odoribacter*	Decreased [174]
33.	*Oscillobacter*	Increased/Decreased [19,180]
34.	*Oscillospira*	Decreased [176]
35.	*Parabacteroides*	Increased [83]
36.	*Peptoniphilus*	Increased [177]
37.	*Porphyromonas*	Increased [43]
38.	*Prevotella*	Increased/Decreased [83,178]
39.	*Pseudobutyrivbrio*	Decreased [177]
40.	*Robinsoniella*	Increased [184]
41.	*Roseburia*	Increased [83]
42.	*Ruminococcus*	Increased/Decreased [176,179]
43.	*SR1 genera incertae sedis*	Decreased [182]
44.	*Streptococcus*	Increased [83]
45.	*Subdoligranulum*	Decreased [19]
46.	*Shigella*	Increased [183]

* Including NAFLD subtypes, namely NAFL, NASH, and NAFLD-related advanced fibrosis.

## Data Availability

All data are inserted in the manuscript and retrievable from PubMed, Scopus, and Google platforms. The first and senior authors are available to share more interpretations and data on request.

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
