# Peer review of "Double Trouble: How Microbiome Dysbiosis and Mitochondrial Dysfunction Drive Non-Alcoholic Fatty Liver Disease and Non-Alcoholic Steatohepatitis"

_biomedicines, 2024, doi:10.3390/biomedicines12030550_

Round 1
Reviewer 1 Report
Comments and Suggestions for Authors
I really enjoyed reviewing this paper. I have only some few comments:
1) The quality of the figures should be improved....maybe even using color figures would help
2) The authors should add some clinica implications on this topic and they should mention some general concepts on preventive measures of MAFLD-related HCC (in this regar cite the recent SRMA: PMID: 33721336 )
3) Maybe the review is too long, particularly in the first part....
Author Response
Thank you for your valuable feedback and for providing specific comments to improve our manuscript. We appreciate the opportunity to address your concerns and have made the following revisions:
Reviewer 1:
- Figures have been improved by using the Canava tools and graphic design platform.
- PIMD: 33721336 has been cited in our manuscript in a suitable paragraph.
- We agree that the review is long as we try to cover the most recent studies on this topic, and we have setup a multinational platform to give a truly comprehensive review on this topic.
We believe these revisions address your concerns and significantly enhance the transparency and reproducibility of our study. We hope these changes meet your expectations, and we appreciate your guidance in improving our manuscript.
Once again, we would like to express our gratitude for your time and valuable feedback. Your comments have undoubtedly strengthened our manuscript, and we look forward to the possibility of having our revised work considered for publication in your esteemed journal.
Sincerely,
Reviewer 2 Report
Comments and Suggestions for Authors
The authors present a review regarding the connection between the development of non-alcoholic fatty liver disease and changes in the intestinal microbiota, as well as disturbances in the mitochondrial apparatus of cells. The literature review covers a huge number of publications, provides comprehensive information on the topic presented, and therefore deserves attention and will be of interest to readers of the Journal. The presented review corresponds to the topic of the Journal. Despite the obvious advantages, a number of comments can be made.
In the very first sentence, the authors state that the liver is the largest endocrine organ. However, the liver also has a pronounced exocrine function, and it is difficult to say which one prevails in the liver.
In my opinion, many sections of the article duplicate each other, in connection with this the volume of the article is greatly expanding, and some sections are presented in only 2 sentences. In this regard, I would like to recommend that the authors formulate their thoughts more clearly and remove side topics.
It is also necessary to draw some conclusions for each section. In its present form, the authors simply list the data obtained, without indicating what consequences this has. This primarily concerns sections on treatment by influencing the microbiota.
It is necessary to try to draw a conclusion for the entire article.
Author Response
Thank you for your valuable feedback and for providing specific comments to improve our manuscript. We appreciate the opportunity to address your concerns and have made the following revisions:
Reviewer 2:
- The first sentence has been edited.
- A conclusion has been written to summarize all aspects of the review.
We believe these revisions address your concerns and significantly enhance the transparency and reproducibility of our study. We hope these changes meet your expectations, and we appreciate your guidance in improving our manuscript.
Once again, we would like to express our gratitude for your time and valuable feedback. Your comments have undoubtedly strengthened our manuscript, and we look forward to the possibility of having our revised work considered for publication in your esteemed journal.
Sincerely,
Reviewer 3 Report
Comments and Suggestions for Authors
THe Authors should discuss the importance of endotoxin and oxidant stress, previously related to the pathophysiology of NAFLD. They should dedicate a new paragraph to the systemic markers of oxidant stres, such as serum sp-NOX2 andurinary 8-iso-PGF2 alpha.
Comments on the Quality of English LanguageModerate editing of English language is required.
Author Response
Thank you for your valuable feedback and for providing specific comments to improve our manuscript. We appreciate the opportunity to address your concerns and have made the following revisions:
Reviewer 3:
- Paragraphs demonstrating “the role of systemic endotoxin and oxidant stress, such as serum sp-NOX2 and urinary 8-iso-PGF2 alpha, in the pathophysiology of NAFLD” have been included.
We believe these revisions address your concerns and significantly enhance the transparency and reproducibility of our study. We hope these changes meet your expectations, and we appreciate your guidance in improving our manuscript.
Once again, we would like to express our gratitude for your time and valuable feedback. Your comments have undoubtedly strengthened our manuscript, and we look forward to the possibility of having our revised work considered for publication in your esteemed journal.
Sincerely,
Reviewer 4 Report
Comments and Suggestions for Authors
1. NAFLD is an implication of many risk factors, most notably metabolic syndrome, which are not related to dysbiosis of the gut microbiota. However, the gut microbiota products activate inflammasome pathways in the liver, triggering liver injury and deterioration of NAFLD condition. Disruption of the gut barrier together with dysbiosis of gut microbiota is correlated with the severity of steatosis and fibrosis condition. Moreover, most studies have remained at the animal stage, and how differences in gut bacterial species between animals and humans affect microbial metabolism remains unclear.
2. Recently there have been many related review papers published. The content of this manuscript is generally similar to other review papers and does not put forward new ideas or perspectives to plan the direction of future research. Therefore, the contribution of this manuscript is limited.
3. No conclusion section was found in this manuscript.
Author Response
Thank you for your valuable feedback and for providing specific comments to improve our manuscript. We appreciate the opportunity to address your concerns and have made the following revisions:
Reviewer 4:
- A conclusion has been written to summarize all aspects of the review.
We believe these revisions address your concerns and significantly enhance the transparency and reproducibility of our study. We hope these changes meet your expectations, and we appreciate your guidance in improving our manuscript.
Once again, we would like to express our gratitude for your time and valuable feedback. Your comments have undoubtedly strengthened our manuscript, and we look forward to the possibility of having our revised work considered for publication in your esteemed journal.
Sincerely,
Round 2
Reviewer 1 Report
Comments and Suggestions for Authors
The revised version of the manuscript is OK. Thank you!
Author Response
Thank you very much.
Reviewer 2 Report
Comments and Suggestions for Authors
All questions have been satisfactorily answered.
Author Response
Thank you very much.
Reviewer 3 Report
Comments and Suggestions for Authors
The Authors answered correctly to all my queries.
Comments on the Quality of English LanguageModerate editing of English naguage is required.
Author Response
Thank you very much.
Reviewer 4 Report
Comments and Suggestions for Authors
This manuscript has been greatly improved and conclusions section has been added, so I don't have other question.
Author Response
Thank you very much.